# Combining Counting Processes and Classification Improves a Stopping Rule for Technology Assisted Review

**Reem Bin-Hezam**[α,β] and **Mark Stevenson**[α]

[α]Department of Computer Science, University of Sheffield, United Kingdom
[β]Information Systems Department, Princess Nourah bint Abdulrahman University, Saudi Arabia
rybinhezam@pnu.edu.sa , mark.stevenson@sheffield.ac.uk

## Abstract

Technology Assisted Review (TAR) stopping rules aim to reduce the cost of manually assessing documents for relevance by minimising the number of documents that need to be examined to ensure a desired level of recall. This paper extends an effective stopping rule using information derived from a text classifier that can be trained without the need for any additional annotation. Experiments on multiple data sets (CLEF e-Health, TREC Total Recall, TREC Legal and RCV1) showed that the proposed approach consistently improves performance and outperforms several alternative methods.[1]

## 1 Background

Information Retrieval (IR) systems often return large numbers of documents in response to user queries and screening them for relevance represents a significant effort. This problem is particularly acute in applications where it is important to identify most (or all) of the relevant documents, for example, systematic reviews of scientific literature (Higgins et al., 2019) and legal eDiscovery (Oard et al., 2018). Technology Assisted Review (TAR) develops methods that aim to reduce the effort required to screen a collection of documents for relevance, such as stopping rules that inform reviewers that a desired level of recall has been reached (and no further documents need to be examined) (Li and Kanoulas, 2020; Yang et al., 2021a).

One of the most commonly applied approaches to developing stopping rules is to identify when a particular level of recall has been reached by estimating the total number of documents in the collection (either directly or indirectly). The majority of algorithms using this approach operate by obtaining manual judgements for a sample of the unobserved documents and inferring the total number of relevant documents, e.g. (Shemilt et al., 2014; Howard et al., 2020; Callaghan and Müller-Hansen, 2020). However, these approaches may not account for the fact that most relevant documents appear early in the ranking, information shown to be useful for stopping algorithms (Cormack and Grossman, 2016a; Li and Kanoulas, 2020), so they may examine more documents than necessary.

**Counting Processes** Counting processes, stochastic models of the number of occurrences of an event within some interval (Cox and Isham, 1980), can naturally model changes to the frequency of relevant document occurances within a ranking. They have been used to develop stopping rules that ensure that a desired level of recall is reached while also minimising the number of documents that need to be reviewed and found to outperform a range of alternative approaches (Sneyd and Stevenson, 2019, 2021; Stevenson and Bin-Hezam, to appear). However, in these methods the estimate of the total number of relevant documents is only based on the examined initial portion of the ranking and no information is used from the remaining documents.

**Classification** Another approach has been to use a text classifier to estimate the total number of relevant documents (Yu and Menzies, 2019; Yang et al., 2021a). This method has the advantage of using information about all documents in the ranking by observing the classifier's predictions for the documents that have not yet been examined. However, the classifier alone does not consider or model the occurrence rate at which relevant documents have already been observed. These methods were found to be effective, although each was only tested on a single dataset.

This paper proposes an extension to stopping algorithms based on counting processes by using a text classifier to inform the estimate of the total number of relevant documents. This approach makes use of information about the relevance of documents from the entire ranking without increas-

---

ing the number of documents that need to be examined for relevance, since the text classifier is trained using information already available.

## 2 Approach

We begin by describing the existing counting process approach and then explain how the classifier is integrated.

**Counting Process Stopping Rule** (Sneyd and Stevenson, 2019, 2021) The approach starts by obtaining relevance judgements for the $n$ highest ranked documents. A rate function is then fitted to model the probability of relevant documents being encountered in the (as yet) unexamined documents later in the ranking. The counting process uses this rate function to estimate the total number of relevant documents remaining. Based on this estimate, the algorithm stops if enough relevant documents have been found to reach a desired recall. If not then more documents are examined and the process repeated until either the stopping point is reached or all documents have been examined. Fitting an appropriate rate function is an important factor in the success of this approach. A set of sample points are extracted from the documents for which the relevance judgements are available (i.e. top $n$), and the probability of a document being relevant at each point is estimated. This set of points is then provided to a non-linear least squares curve fitting algorithm to produce the fitted rate function. For each sample point, the probability of a document being relevant is computed by simply examining a window of documents around it and computing the proportion that is relevant, i.e.

$$\frac{1}{n} \sum_{d \in W} \mathbb{1}(d) \qquad (1)$$

where $W$ is the set of documents in the window and $\mathbb{1}(d)$ returns 1 if $d$ has been labelled as relevant.

**Integrating Text Classification** Since the existing counting process approach only examines the top-ranked documents, it relies on the assumption that the examined portion of documents provides a reliable indication of the rate at which relevant documents occur within the ranking. However, this may not always be the case, particularly when only a small number of documents have been manually examined or when relevant documents unexpectedly appear late in the ranking. This problem can be avoided by applying a text classifier to the unexamined documents at each stage. The information

it provides helps to ensure that the rate function is appropriate and that the estimate of the total number of relevant documents provided by the counting process is accurate.

The text classifier is integrated into the stopping algorithm in a straightforward way. We begin by assuming that the ranking contains a total of $N$ documents. As before, relevance judgements are obtained for the first $n$. These judgements are used to train a text classifier which is applied to all remaining documents without relevance judgements (i.e. $n+1 \ldots N$). The rate function is now fitted by examining all $N$ documents in the ranking (rather than only the first $n$), using manual relevance judgements for the first $n$ and the classifier's predictions for the remainder (i.e. $n+1 \ldots N$). As before, the algorithm stops if it has been estimated that enough relevant documents are contained within the first $n$ to achieve the desired recall. Otherwise, the value of $n$ is increased and the process repeated. See Figure 1.

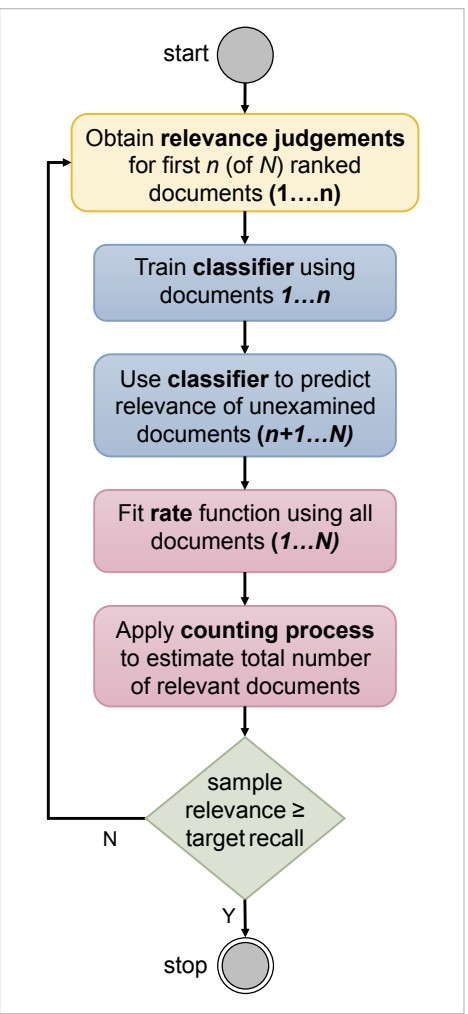

Figure 1: Counting Process with Classifier Approach

It is possible to use information from the classifier in various ways when computing the probability of document relevance during the curve fitting process. Two were explored:

**ClassLabel**: the class labels output by the classifier are used directly, i.e. using eq. 1 with relevance determined by the classifier.

**ClassScore**: Alternatively, the scores output by the classifier is used, subject to a threshold of 0.5, i.e. eq. 1 is replaced by

$$\frac{1}{n} \sum_{d \in W} \begin{cases} score(d) & \text{if } score(d) \geq 0.5 \\ 0 & \text{otherwise} \end{cases} \quad (2)$$

where $score(d)$ is the score the classifier assigned to $d$. (Using the class score without thresholding was found to overestimate the number of relevant documents.)

## 3 Experiments

Experiments compared three approaches. **CP**: Baseline counting process approach without text classification, i.e. (Sneyd and Stevenson, 2021). **CP+ClassLabel** and **CP+ClassScore**: Counting process combined with the text classifier (see §2).

The counting process uses the most effective approach previously reported (Sneyd and Stevenson, 2021). A power law (Zobel, 1998) was used as the rate function with an inhomogeneous Poisson process since its performance is comparable with a Cox process while being more computationally efficient. Parameters were set as follows: the confidence level was set to 0.95 while the sample size was initialised to 2.5% and incremented by 2.5% for each subsequent iteration. An existing reference implementation was used (Sneyd and Stevenson, 2021).

The text classifier uses logistic regression which has been shown to be effective for TAR problems (Yang et al., 2021a; Li and Kanoulas, 2020). The classifier was based on scikit-learn using TF-IDF scores of each document's content as features, the content is title and abstract for CLEF datasets, email messages for TREC datasets (including title and message body), and news articles content for the RCV1 dataset. The classification threshold was set to 0.5 (model default) by optimising F-measure for relevant documents using a held-out portion of the dataset. Text classifiers based on BioBERT (Lee et al., 2020) and PubMedBERT (Gu et al., 2020) were also developed. However, neither outperformed the linear logistic regression model on

CLEF datasets, a result consistent with previous findings for TAR models (Yang et al., 2022; Sadri and Cormack, 2022).

TAR problems are often highly imbalanced, with very few relevant documents. However, TAR methods successfully place relevant documents early in the ranking (Cormack and Grossman, 2015, 2016b; Li and Kanoulas, 2020). Consequently, the prevalence of relevant documents in the training data is unlikely to be representative of the entire collection, particularly early in the ranking. To account for this cost-sensitive learning was used during classifier training. The weight of the minority class within the training data was set to 1, while the weight of the majority class was set to the imbalance ratio (i.e. count of the minority over the majority class in training data) (Ling and Sheng, 2008).

### 3.1 Datasets

Performance was evaluated on six diverse datasets widely used in previous work on TAR. **CLEF e-Health (CLEF2017, CLEF2018, CLEF2019)** (Kanoulas et al., 2017, 2018, 2019): A collection of systematic reviews from the Conference and Labs of the Evaluation Forum (CLEF) 2017, 2018, and 2019 e-Health lab Task 2: Technology-Assisted Reviews in Empirical Medicine. The datasets contain 42, 30, and 31 reviews respectively. **TREC Total Recall (TR)** (Grossman et al., 2016) A collection of 290,099 emails with 34 topics related to Jeb Bush's eight-year tenure as Governor of Florida (athome4). **TREC Legal (Legal)** (Cormack et al., 2010) A collection of 685,592 Enron emails made available by the Federal Energy Review Commission during their investigation into the company's collapse. **RCV1** (Lewis et al., 2004) A collection of Reuters news articles labelled with categories. Following previous work (Yang et al., 2021b,a), 45 categories were used to represent a range of topics, and the collections downsampled to 20%.

The ranking used by the stopping model needs to be the same for all datasets used in the experiments to ensure fair and meaningful comparison. The rankings produced by AutoTAR (Cormack and Grossman, 2015) were used since they allow comparison with a range of alternative approaches (Li and Kanoulas, 2020) and can be generated using their reference implementation.

### 3.2 Evaluation Metrics

Approaches were evaluated using the metrics commonly used in previous work on TAR stopping

Table 1: Results for Counting Process and Classification Stopping Methods. (* indicate scores are *Not* statistically significantly different from CP, + indicate CP+ClassLabel and CP+ClassScore are statistically significantly different)

| Dataset | Model | Desired recall = 0.9 | | | | Desired recall = 0.8 | | | | Desired recall = 0.7 | | | |
|---|---|---|---|---|---|---|---|---|---|---|---|---|---|
| | | Recall | Rel. | Cost | Excess | Recall | Rel. | Cost | Excess | Recall | Rel. | Cost | Excess |
| CLEF2017 | CP | 1.000 | **1.000** | 0.281 | 0.238 | 1.000 | **1.000** | 0.265 | 0.232 | 1.000 | **1.000** | 0.255 | 0.229 |
| | CP+ClassLabel | **0.989** | **1.000** | 0.153 | 0.102 | **0.989** | **1.000** | **0.152** | 0.114 | **0.988** | **1.000** | 0.150 | 0.120 |
| | CP+ClassScore | **0.989** | **1.000** | **0.152** | **0.101** | **0.989** | **1.000** | **0.152** | 0.114 | **0.988** | **1.000** | **0.147** | **0.117** |
| CLEF2018 | CP | 1.000 | **1.000** | 0.293 | 0.242 | 1.000 | **1.000** | 0.287 | 0.249 | 1.000 | **1.000** | 0.277 | 0.245 |
| | CP+ClassLabel | **0.983** | **1.000** | **0.137** | **0.075** | 0.983 | **1.000** | 0.137 | 0.091 | 0.982 | **1.000** | 0.135 | 0.097 |
| | CP+ClassScore | **0.983** | **1.000** | **0.137** | **0.075** | **0.982** | **1.000** | **0.136** | **0.090** | **0.981** | **1.000** | **0.134** | **0.096** |
| CLEF2019 | CP | 0.999 | **1.000** | 0.283 | 0.228 | 0.999 | **1.000** | 0.279 | 0.235 | 0.999 | **1.000** | 0.276 | 0.240 |
| | CP+ClassLabel | **0.996** | **1.000** | 0.221 | 0.161 | **0.996** | **1.000** | 0.216 | 0.169 | **0.996** | **1.000** | 0.212 | 0.173 |
| | CP+ClassScore | **0.996** | **1.000** | 0.221 | 0.161 | **0.996** | **1.000** | **0.213$^+$** | **0.165$^+$** | **0.994** | **1.000** | **0.207$^+$** | **0.168$^+$** |
| Legal | CP | 1.000 | **1.000** | 0.425 | 0.401 | 1.000 | **1.000** | 0.338 | 0.320 | 1.000 | **1.000** | 0.287 | 0.273 |
| | CP+ClassLabel | **0.972*** | **1.000** | **0.088** | **0.050** | **0.972*** | **1.000** | **0.088** | 0.064 | **0.972*** | **1.000** | 0.088 | 0.070 |
| | CP+ClassScore | **0.972*** | **1.000** | **0.088** | **0.050** | **0.972*** | **1.000** | **0.088** | 0.064 | **0.963*** | **1.000** | **0.075** | **0.057** |
| TR | CP | 1.000 | **1.000** | 0.059 | 0.054 | 1.000 | **1.000** | 0.056 | 0.052 | 1.000 | **1.000** | 0.052 | 0.049 |
| | CP+ClassLabel | **0.999** | **1.000** | **0.028** | **0.023** | **0.999*** | **1.000** | **0.027** | **0.023** | **0.999*** | **1.000** | **0.027** | 0.024 |
| | CP+ClassScore | **0.999** | **1.000** | **0.028** | **0.023** | **0.999*** | **1.000** | **0.027** | **0.023** | **0.999*** | **1.000** | **0.027** | 0.024 |
| RCV1 | CP | 0.999 | **1.000** | 0.193 | 0.180 | 0.998 | **1.000** | 0.154 | 0.145 | 0.998 | **1.000** | 0.134 | 0.127 |
| | CP+ClassLabel | 0.972 | 0.956* | 0.038 | 0.022 | **0.969** | **1.000** | 0.036 | 0.026 | **0.969** | **1.000** | 0.036 | 0.028 |
| | CP+ClassScore | **0.969** | 0.933* | **0.036** | **0.020** | **0.969** | **1.000** | **0.036** | 0.026 | **0.969** | **1.000** | 0.036 | 0.028 |

criteria, e.g. (Yang et al., 2021a; Li and Kanoulas, 2020), and calculated using the `tar_eval` open-source evaluation script.[2] Average scores across all topics in each collection are reported.

**Recall**: proportion of relevant documents identified by the method. Following Li and Kanoulas (2020), results closest to the desired recall are considered best, rather than the highest overall recall.

**Reliability (Rel.)**: percentage of topics where the desired recall was reached (or exceeded). For each topic, reliability is 1 if the desired recall was reached, and 0 otherwise.

**Cost**: percentage of documents examined.

**Excess cost (Excess)**: we introduce this measure which quantifies the additional documents that have to be examined compared with optimal stopping (i.e. stopping immediately when the desired recall has been reached). It is computed as follows:

$$excess\ cost = \frac{cost(method) - cost(optimal)}{1 - cost(optimal)}$$
(3)

where $cost(method)$ and $cost(optimal)$ are the cost of the method being evaluated and stopping at the optimal point, respectively. This metric indicates the proportion of the documents that need to be examined after the desired recall has been reached.

[2] https://github.com/CLEF-TAR/tar

## 4 Results and Analysis

Experiments were carried out using a range of desired recalls {0.9, 0.8, 0.7} with results shown in Table 1. Results show that combining the classifier with the counting process (CP+ClassLabel and CP+ClassScore) is more effective than using the counting process alone. The improvement in the performance of both approaches compared with CP is statistically significant across topics ($p < 0.05$, paired t-test with Bonferroni correction). There is a significant reduction in cost which is often substantial (e.g. Legal collection with a desired recall of 0.9). This is achieved with a minimal reduction to the number of relevant documents identified, although the reliability remains unaffected in the majority of cases, and when it is affected (RCV1 collection with a desired recall of 0.9), the reduction is minimal and not statistically significant.

The performance of CP+ClassScore and CP+ClassLabel are comparable. The scores for CP+ClassScore may be marginally better, although the difference is only significant in limited circumstances. (Differences in the cost and excess scores were significant for the CLEF 2019 collection with desired recalls of 0.8 and 0.7.) Overall, the way in which the classifier output is integrated into the approach seems less important than the fact it is used at all. The average recall for all approaches exceeds the desired recall, indicating a tendency to be somewhat conservative in proposing the stopping

Table 2: Proposed Method vs. Baselines (TR collection)

| Model | Desired recall = 0.9 | | | | Desired recall = 0.8 | | | |
|---|---|---|---|---|---|---|---|---|
| | Recall | Rel. | Cost | Excess | Recall | Rel. | Cost | Excess |
| CP+ClassScore | 0.999 | **1.000** | **0.028** | **0.023** | 0.999 | **1.000** | **0.027** | **0.023** |
| SCAL (Cormack and Grossman, 2016b) | 0.903 | 0.647 | 0.144 | 0.140 | 0.761 | 0.676 | 0.107 | 0.103 |
| SD-training (Hollmann and Eickhoff, 2017) | 1.000 | **1.000** | 1.000 | 1.000 | 1.000 | **1.000** | 1.000 | 1.000 |
| SD-sampling (Hollmann and Eickhoff, 2017) | 0.936 | 0.794 | 0.779 | 0.778 | 0.896 | 0.735 | 0.690 | 0.689 |
| AutoStop (Li and Kanoulas, 2020) | 0.953 | 0.941 | 0.766 | 0.765 | 0.885 | 0.912 | 0.754 | 0.753 |
| Target (Cormack and Grossman, 2016a) | **0.900** | 0.706 | 0.069 | 0.064 | **0.844** | 0.882 | 0.069 | 0.065 |

point. However, the classifier allows the algorithm to identify this point earlier, presumably because it has indicated a low probability of relevant documents being found later in the ranking.

## 4.1 Effect of Cost-sensitive Learning

The effect of using cost-sensitive learning during classifier training was explored by comparing it with the performance obtained when it was not used, see Table 3. These results demonstrate the importance of accounting for class imbalance when training the classifier. The low prevalence of relevant documents can lead to the classifier being unable to identify them resulting in the approach stopping too early, particularly when the desired recall is high.

Table 3: Effect of Cost-Sensitive Learning (CSL) for CP+ClassLabel Model applied to Legal collection

| Desired recall | CSL | Recall | Rel. | Cost | Excess |
|---|---|---|---|---|---|
| 0.9 | ✗ | 0.793 | 0.000 | 0.025 | -0.016 |
| | ✓ | 0.972 | 1.000 | 0.088 | 0.050 |
| 0.8 | ✗ | 0.793 | 0.500 | 0.025 | -0.001 |
| | ✓ | 0.972 | 1.000 | 0.088 | 0.064 |
| 0.7 | ✗ | 0.793 | 1.000 | 0.025 | 0.006 |
| | ✓ | 0.972 | 1.000 | 0.088 | 0.070 |

## 4.2 Baseline Comparison

Direct comparison of stopping methods is made difficult by the range of rankings and evaluation metrics used in previous work. However, Li and Kanoulas (2020) reported the performance of several approaches using the same rankings as our experiments, allowing benchmarking against several methods. SCAL (Cormack and Grossman, 2016b), and AutoStop (Li and Kanoulas, 2020) sample documents from the entire ranking to estimate the number of relevant documents, SD-training/SD-sampling (Hollmann and Eickhoff, 2017) use the scores assigned by the ranking. Results for the

target method (Cormack and Grossman, 2016a) are also reported. Table 2 compares the proposed approach against these baselines for the TR collection for the desired recall levels for which results are available. (Results for other datasets were similar and available in Li and Kanoulas (2020).) CP+ClassScore has a lower cost than the baselines, requiring only a fraction of the documents to be examined, while achieving a higher recall. Although the average recall of some baselines is closer to the desired recall than the proposed approach, their reliability is also lower which indicates that they failed to identify enough relevant documents to achieve the desired recall for multiple topics.

## 5 Conclusion

This paper explored the integration of a text classifier into an existing TAR stopping algorithm. Experiments on six collections indicated that the proposed approach was able to achieve the desired recall level with a statistically significant lower cost than the existing method based on counting processes alone. Integrating the classifier output in the form of labels or scores made little difference, with improvements observed using either approach. The text classifier provides the stopping algorithm with information about the likely relevance of the documents that have not yet been manually examined, allowing it to better estimate the number remaining.

## Limitations

We presented a stopping rule for ranked lists. Comparison against multiple datasets requires the same ranking model for all datasets, which may not always be available. In addition, reported results have been limited to a single rate function (power law) and confidence level (0.95). However, the pattern of results for other hyperparameters is similar to those reported.

## Ethics Statement

This work presents a TAR method designed to reduce the amount of effort required to identify information of interest within large document collections. The most common applications of this technology are literature reviews and identification of documents for legal disclosure. We do not foresee any ethical issues associated with these uses.

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
