# OpenReview forum: "Combining Counting Processes and Classification Improves a Stopping Rule for Technology Assisted Review"
_EMNLP/2023/Conference — EMNLP 2023 Findings_

### Official Review · Reviewer_jZAo · 2023-08-03

**Soundness:** 3

**Excitement:**

3: Ambivalent: It has merits (e.g., it reports state-of-the-art results, the idea is nice), but there are key weaknesses (e.g., it describes incremental work), and it can significantly benefit from another round of revision. However, I won't object to accepting it if my co-reviewers champion it.

**Paper Topic And Main Contributions:**

In the context of information retrieval, many documents are often returned. Therefore, it is difficult to analyze all documents, but it is important to identify the most relevant documents depending on the use case. The paper presents a method in the context of technology-assisted review (TAR) to reduce the number of relevant documents. Related work presented a stopping rule based on counting processes. This work extends this counting process by using a text classifier.

Contributions:
- a method to integrate text classification into the stopping rule in the context of technology-assisted review/information retrieval
- testing of a novel approach on a large range of different datasets
- showed the benefit of their model


**Questions For The Authors:**

There are some aspects unclear:

- In line 114 you write 'if d has been labelled as relevant', and then in line 122 you say that the counting process might not provide reliable results when e.g. only a small number of documents have been manually examined. So how do you define in your setup something as relevant? Is this a manual process in general? Somebody would analyse the first set of documents and label them? And in your case, would this be given, as the data is already labelled?

- About the classifier: You use the first n samples with their labels and train a classifier on them. Generally, it would be interesting to know how well the classifier performs and how it performs then on the unexamined documents. I would assume that the label distribution changes between examined to unexamined documents, which might strongly influence the classifier's performance. Also, is there some kind of correlation between your approach's performance and the classifier's reliability? These would certainly be nice aspects to explore in more detail, to improve the work....

- Excess cost evaluation is not clear to me. Can you say a bit more about cost(method) and cost(optimal)?


**Reasons To Accept:**

- The paper describes a small focussed contribution. Therefore it fits as a short paper.
- At the same time, the authors test their approach on a large variety of different datasets and show that it works for their setup.

**Reasons To Reject:**

- A more meaningful comparison might be good. The work is just compared to one approach (except for one dataset).
- A more detailed analysis which provides more insights (e.g. in the context of the classifier), might be very valuable.

**Reproducibility:**

3: Could reproduce the results with some difficulty. The settings of parameters are underspecified or subjectively determined; the training/evaluation data are not widely available.

**Reviewer Confidence:**

2: Willing to defend my evaluation, but it is fairly likely that I missed some details, didn't understand some central points, or can't be sure about the novelty of the work.

**Typos Grammar Style And Presentation Improvements:**

- There might be an error in reference of Sadri and Cormack, 2022? Please check the reference part at the end.
- line 248: This should probably start with a capital 'e'?

---

> ### Author Rebuttal · Authors · 2023-08-29
>
> We thank the reviewers for their constructive comments and feedback on our work. We want to address their questions and comments below.
>
>
> **Questions For The Authors:**
>
>
> * **3A: In line 114 you write 'if d has been labelled as relevant', and then in line 122 you say that the counting process might not provide reliable results when e.g. only a small number of documents have been manually examined. So how do you define in your setup something as relevant? Is this a manual process in general? Somebody would analyse the first set of documents and label them? And in your case, would this be given, as the data is already labelled?**
>
> * Yes, the examined documents are already labelled by human experts and the datasets we used are already labelled. However, while using the classifier, we are training the classifier on the examined set of documents with true labels, and apply it to the unexamined part to get the classifier predictions (labels or scores). Therefore, for the examined part relevance is labelled by human experts, and for the unexamined part, relevance is predicted by the classifier.
>
> ---
>
> * **3B (1): About the classifier: You use the first n samples with their labels and train a classifier on them. Generally, it would be interesting to know how well the classifier performs and how it performs then on the unexamined documents.**
>
> * Classifier performance improved as more documents are included in the training set at each iteration. Without any imbalance handling, the initial classifier performance was biased towards the majority class, which was fixed later with proper imbalance handling tehcnique (cost-sensitve learning in our experiments).
>
> ---
>
> * **3B (2): I would assume that the label distribution changes between examined to unexamined documents, which might strongly influence the classifier's performance.**
> * Yes the label distribution changes between examined and unexamined documents, since TAR problems are usually highly imbalanced, that's why we handled this using cost-sensitive learning. With ranked list, the majority of the training dataset would be relevant, while the majority of the test set would be non-relevant. The training data changes at each iteration as more examined documents are included, and thus the imbalance ratio is also updated and re-calculated based on the training dataset majority/minority class at each iteration.
>
> ---
>
> * **3B (3): Also, is there some kind of correlation between your approach's performance and the classifier's reliability? These would certainly be nice aspects to explore in more detail, to improve the work....**
>
> * Yes, definitely the stopping decision is affected by the classifier's performance. We have explored the classifier performance without using any imbalance handling, which causes the learning to be biased toward the majority class at each iteration, and the stopping decision would be either too early (causes lower reliability as low as 0 for some datasets), where the majority in the training was the non-relevant class, or the stopping would be too late (cause higher cost), where the majority in the training was the relevant class.
>
> * However, applying the cost-sensitive learning improved the classifier's performance and hence the stopping decision became more accurate. Handling the imbalance with cost-sensitive learning was selected over using oversampling techniques because, although they are a good approach, in some cases where the minority class is too small in the sample, the generated synthetic instances by SMOTE are not going to be representative of the class, and hence results would not be accurate. Moreover, cost-sensitive was selected over using undersampling because it may cause the loss of some potentially useful instances of the relevant class during the learning phase, since labelling relevant documents correctly and not missing any of them by the classifier is important for many TAR problems.
>
> * The table below shows the results of how the classifier performance drops while not using any imbalance handling techniques, which affects the overall stopping decisions and causes lower reliability and undershooting of the desired recall.
>
> Table 1: CP+ClassLabel Model Results with Original Data without any Imbalance Handling
>
> | dataset  | des_recall | recall | reliability | cost  | excess |
> |----------|------------|--------|-------------|-------|--------|
> | CLEF2017 | 0.9        | 0.967  | 0.867       | 0.133 | 0.081  |
> |          | 0.8        | 0.967  | 0.967       | 0.133 | 0.094  |
> |          | 0.7        | 0.966  | 1           | 0.131 | 0.100  |
> | Legal    | 0.9        | 0.793  | 0           | 0.025 | -0.016 |
> |          | 0.8        | 0.793  | 0.5         | 0.025 | -0.001 |
> |          | 0.7        | 0.793  | 1           | 0.025 | 0.006  |
>
>
> ---
>
> * **3C: Excess cost evaluation is not clear to me. Can you say a bit more about cost(method) and cost(optimal)?**
>
> * cost(method) is the cost of the method being evaluated (e.g. CP+ClassLabel), and cost(optimal) is the cost of stopping at the optimal point once the desired recall is achieved. The cost here is calculated as the percentage of examined documents from the whole collection.
> * The Excess metric indicates the proportion of the documents that are examined after the desired recall has been reached (positve: overshooting), or that need to be examined until the desired recall cost is reached (negative: undershooting).
>
> ---
>
> **Typos Grammar Style And Presentation Improvements:**
>
> Thanks a lot, we’ll fix this.
>
>
> ---
>
>
> **Other Comments**
>
> * Results for baselines with all CLEF & TREC datasets are already available but were not included because of the space limit. They follow the same pattern and could be included when the extra page is provided. The table below summarises our approaches with the baselines on more datasets.
>
> Table 2: Proposed Methods vs. Baseline
>
> |          |               | des_recall = 0.9 |             |       |        | des_recall = 0.8 |             |       |        |
> |----------|---------------|:----------------:|:-----------:|:-----:|:------:|:----------------:|:-----------:|:-----:|:------:|
> | Dataset  | model         | recall           | reliability | cost  | excess | recall           | reliability | cost  | excess |
> | CLEF2017 | CP            |            1.000 |       1.000 | 0.281 |  0.238 |            1.000 |       1.000 | 0.265 |  0.232 |
> |          | CP+ClassLabel |            0.989 |       1.000 | 0.153 |  0.102 |            0.989 |       1.000 | 0.152 |  0.114 |
> |          | CP+ClassScore |            0.989 |       1.000 | 0.152 |  0.101 |            0.989 |       1.000 | 0.152 |  0.114 |
> |          | SCAL          |            0.914 |       0.667 | 0.496 |  0.466 |            0.888 |       0.800 | 0.451 |  0.426 |
> |          | SD-training   |            0.955 |       0.833 | 0.691 |  0.672 |            0.881 |       0.767 | 0.417 |  0.391 |
> |          | SD-sampling   |            0.902 |       0.567 | 0.506 |  0.476 |            0.798 |       0.433 | 0.350 |  0.321 |
> |          | AutoStop      |            0.884 |       0.500 | 0.421 |  0.386 |            0.787 |       0.367 | 0.335 |  0.305 |
> |          | Target        |            0.960 |       1.000 | 0.498 |  0.468 |            0.922 |       1.000 | 0.468 |  0.444 |
> | Dataset  | model         | recall           | reliability | cost  | excess | recall           | reliability | cost  | excess |
> | CLEF2018 | CP            |            1.000 |       1.000 | 0.293 |  0.242 |            1.000 |       1.000 | 0.287 |  0.249 |
> |          | CP+ClassLabel |            0.983 |       1.000 | 0.137 |  0.075 |            0.983 |       1.000 | 0.137 |  0.091 |
> |          | CP+ClassScore |            0.983 |       1.000 | 0.137 |  0.075 |            0.982 |       1.000 | 0.136 |  0.090 |
> |          | SCAL          |            0.902 |       0.667 | 0.493 |  0.457 |            0.862 |       0.767 | 0.428 |  0.397 |
> |          | SD-training   |            0.972 |       0.967 | 0.701 |  0.680 |            0.886 |       0.833 | 0.414 |  0.383 |
> |          | SD-sampling   |            0.855 |       0.367 | 0.379 |  0.334 |            0.753 |       0.367 | 0.258 |  0.218 |
> |          | AutoStop      |            0.892 |       0.600 | 0.441 |  0.401 |            0.781 |       0.467 | 0.347 |  0.312 |
> |          | Target        |            0.934 |       0.833 | 0.431 |  0.390 |            0.908 |       0.967 | 0.421 |  0.390 |
> | Dataset  | model         | recall           | reliability | cost  | excess | recall           | reliability | cost  | excess |
> | CLEF2019 | CP            |            0.999 |       1.000 | 0.283 |  0.228 |            0.999 |       1.000 | 0.279 |  0.235 |
> |          | CP+ClassLabel |            0.996 |       1.000 | 0.221 |  0.161 |            0.996 |       1.000 | 0.216 |  0.169 |
> |          | CP+ClassScore |            0.996 |       1.000 | 0.221 |  0.161 |            0.996 |       1.000 | 0.213 |  0.165 |
> |          | SCAL          |            0.893 |       0.516 | 0.621 |  0.592 |            0.887 |       0.903 | 0.577 |  0.551 |
> |          | SD-training   |            0.940 |       0.774 | 0.713 |  0.691 |            0.826 |       0.613 | 0.421 |  0.386 |
> |          | SD-sampling   |            0.893 |       0.508 | 0.517 |  0.480 |            0.787 |       0.475 | 0.366 |  0.328 |
> |          | AutoStop      |            0.878 |       0.387 | 0.479 |  0.439 |            0.791 |       0.452 | 0.397 |  0.361 |
> |          | Target        |            0.968 |       0.935 | 0.673 |  0.648 |            0.945 |       1.000 | 0.648 |  0.627 |
> | Dataset  | model         | recall           | reliability | cost  | excess | recall           | reliability | cost  | excess |
> |   Legal  | CP            |            1.000 |       1.000 | 0.425 |  0.401 |            1.000 |       1.000 | 0.056 |  0.052 |
> |          | CP+ClassLabel |            0.972 |       1.000 | 0.088 |  0.050 |            0.999 |       1.000 | 0.027 |  0.023 |
> |          | CP+ClassScore |            0.972 |       1.000 | 0.088 |  0.050 |            0.999 |       1.000 | 0.027 |  0.023 |
> |          | SCAL          |            0.039 |       0.000 | 0.005 | -0.036 |            0.039 |       0.000 | 0.005 |  0.001 |
> |          | SD-training   |            1.000 |       1.000 | 1.000 |  1.000 |            1.000 |       1.000 | 1.000 |  1.000 |
> |          | SD-sampling   |            1.000 |       1.000 | 1.000 |  1.000 |            1.000 |       1.000 | 1.000 |  1.000 |
> |          | AutoStop      |            0.803 |       0.000 | 0.811 |  0.803 |            0.684 |       0.000 | 0.794 |  0.793 |
> |          | Target        |            0.882 |       0.500 | 0.092 |  0.054 |            0.808 |       1.000 | 0.073 |  0.069 |
> | Dataset  | model         | recall           | reliability | cost  | excess | recall           | reliability | cost  | excess |
> |    TR    | CP            |            1.000 |       1.000 | 0.059 |  0.054 |            1.000 |       1.000 | 0.056 |  0.052 |
> |          | CP+ClassLabel |            0.999 |       1.000 | 0.028 |  0.023 |            0.999 |       1.000 | 0.027 |  0.023 |
> |          | CP+ClassScore |            0.999 |       1.000 | 0.028 |  0.023 |            0.999 |       1.000 | 0.027 |  0.023 |
> |          | SCAL          |            0.903 |       0.647 | 0.144 |  0.140 |            0.761 |       0.676 | 0.107 |  0.103 |
> |          | SD-training   |            1.000 |       1.000 | 1.000 |  1.000 |            1.000 |       1.000 | 1.000 |  1.000 |
> |          | SD-sampling   |            0.936 |       0.794 | 0.779 |  0.778 |            0.896 |       0.735 | 0.690 |  0.689 |
> |          | AutoStop      |            0.953 |       0.941 | 0.766 |  0.765 |            0.885 |       0.912 | 0.754 |  0.753 |
> |          | Target        |            0.900 |       0.706 | 0.069 |  0.064 |            0.844 |       0.882 | 0.069 |  0.065 |

---

### Official Review · Reviewer_Mr4c · 2023-08-03

**Soundness:** 3

**Excitement:**

3: Ambivalent: It has merits (e.g., it reports state-of-the-art results, the idea is nice), but there are key weaknesses (e.g., it describes incremental work), and it can significantly benefit from another round of revision. However, I won't object to accepting it if my co-reviewers champion it.

**Paper Topic And Main Contributions:**

This paper deals with information retrieval, in particular for Technology Assisted Review (TAR) settings where high recall is important. The TAR setting considered is one where an iterative process is set up that aims to reduce the work needed to scan a set of documents for relevance to some topic. The iterative process will keep choosing batches of documents that are ranked as most relevant by the AutoTAR system to provide to human annotators to mark as relevant or not until some stopping criterion is met. This paper focuses on how to detect when to stop the TAR process by building on counting process approaches published previously in (Sneyd and Stevenson, 2019, 2021). The counting process stopping methods work by fitting a relevancy rate function to model the chances of relevant documents occurring lower in the ranking. The counting process uses the fitted rate function to estimate the total number of relevant documents remaining in the collection. Based on this estimate, then the process is stopped when a desired level of estimated recall is reached. The main contribution of the paper is to improve the stopping rule by integrating text classification in order to improve the modeling of relevance rates. The intuition is that the counting process approach relies on the assumption that the data from the top-ranked documents is reliable for predicting the rate of relevant documents lower in the ranking, an assumption which may not be true. The paper proposes that a text classifier be built that can classify documents lower in the ranking and then the predicted class labels (named the ClassLabel approach) or class scores thresholded by 0.5 (named the ClassScore approach) on those lower ranked documents can then also be used for fitting the rate function. Experiments are conducted using a power law as the rate function with an inhomogeneous Poisson process and a logistic regression text classifier using tf-idf features. Cost-sensitive learning was used with an imbalance ratio obtained by using the count of the minority class over the count of the majority class in the training data. Experiments were conducted on CLEF e-Health, TREC Total Recall, TREC Legal, and RCV1 datasets. Recall, Reliability, Cost, and excess cost were used as evaluation metrics. Desired recall levels of 0.9, 0.8, and 0.7 were used in the experiments. The proposed integration of text classification improves stopping performance over the existing counting process approach, with the ClassLabel and ClassScore variations not having much difference in performance between each other, but both performing better than the existing counting process approach without text classification integrated. If we use the conservative and aggressive terminology to describe the behavior of the TAR stopping methods in the same sense the terms are used in the active learning stopping methods literature (https://aclanthology.org/W09-1107.pdf), the results of the paper show that the existing counting process stopping methods tend to stop too conservatively and the newly proposed stopping method that integrates the counting process and text classification ideas tends to stop more aggressively, thereby saving annotation effort while not harming recall and reliability.

**Questions For The Authors:**

A: Is it possible to just use the text classification label predictions to estimate recall and avoid the counting processes and curve fitting entirely?   If not, why not? If so, I think it would be beneficial to provide information on how such approaches behave.

B: Lines 198-202 discuss the imbalance ratio and state it is set based on the count of the minority class over the majority class "in training data". Is the training data changing at each iteration of TAR? If so, does this means that the imbalance ratio can change each iteration after more human-provided relevancy labels are provided?  If this is not right, then can you explain how it is set?

**Reasons To Accept:**

A novel method for stopping TAR based on desired levels of recall is proposed. The new method combines ideas from previously published TAR stopping methods that use counting processes and that use text classification.

Experiments cover a range of datasets covering a range of different genres and three different desired recall levels were used. The results seem to relatively consistently show that the new proposed integrated method improves stopping performance for the experimental setup used when compared with using the existing counting process methods.

**Reasons To Reject:**

The new proposed method is not compared with previously published methods that use text classification, such as those methods mentioned on lines 063-065. It's mentioned at lines 288-290 that direct comparison with previous work is difficult due to the range of rankings and evaluation metrics used in previous work. While this could make it difficult to compare performance numbers across different papers, it seems like it should still be feasible to re-implement previous methods and run them directly with the same rankings and evaluation metrics as the new proposed method was run with in the experiments reported. Currently the paper shows results for the counting process approach with no use of text classification and for the combined approach of using counting process with text classification, but it is missing information on how the text classification approaches without using the counting process approach behave.

The paper doesn't seem to dive deeper into understanding what's going on other than that an approach was tried and it improved performance. It seems that the improvement means that the counting process rate fitting was not working so well, which is why there was so much room for improvement, however, there is not any analysis to illuminate why it wasn't working well. It could help to show some of the curves that were fit versus the reality of the observed data to show how the observed data differ from what the rate curves were indicating. From the final results it seems to me that the counting process rules are stopping too conservatively in the sense mentioned earlier in this review, which would seem to imply that they're indicating that stopping should not occur because they're predicting that there are more relevant documents left to be found in the remainder of the collection than there actually are left in the collection. It might be helpful to analyze this overestimation more and shed some light on why this overestimation occurs. It seems possible that functions different from the one power law function that was used in the paper could be worth exploring.

The impact of integrating the text classification component seems like it has the potential to be highly dependent on how the counting process stopping rule is implemented. The paper could be stronger if it experimented with and provided substantial information on the behavior when using different rate functions, confidence levels, sampling of points for curve fitting, window sizes, etc. The paper, to its credit, does mention some of these aspects, although not all of them, as limitations in the limitations section. The mentioning of some of these limitations is better than not mentioning them, however, it does not remove the fact that they remain as limitations.  There is a statement at lines 329-330 that "the pattern of results is similar to those included", but more details would be helpful in clarifying what was tested and what the results were.

**Reproducibility:**

3: Could reproduce the results with some difficulty. The settings of parameters are underspecified or subjectively determined; the training/evaluation data are not widely available.

**Reviewer Confidence:**

4: Quite sure. I tried to check the important points carefully. It's unlikely, though conceivable, that I missed something that should affect my ratings.

**Typos Grammar Style And Presentation Improvements:**

Consider whether it might be more intuitive to replace 'n' with |W| in equation 1.

In general, I sense that the paper might be a little difficult for readers to follow if they don't already know quite a bit about TAR and the (Sneyd and Stevenson, 2019, 2021) papers. Consider if it's possible to make the paper more self-contained.

Figure 1 and Table 2 are too small to read easily when the paper is printed out. Consider if there is a way to make it easier to read from a hard copy of the paper.

There are some spots in the paper where it's possible that providing some more details might be helpful for readers. I found that I think I was able to understand the gist of how the counting process stopping rules work, but I struggled to understand or to find in the paper precise details about how they work. In particular, how are the sample points chosen? For example, are they equally spaced in the relevancy ranking list and systematically chosen, or are they randomly chosen? How many sample points are used? What is the window size W and how is it chosen? Is it fixed or can it vary?

It's mentioned that BioBERT and PubMedBERT models were developed, but that they didn't perform better than the logistic regression model at lines 183-186. It seems an odd choice and a bit confusing as to why one would use BioBERT and PubMedBERT on the TREC Total Recall, TREC Legal, and RCV1 datasets since these datasets don't seem to be from a biomedical domain. Some explanation of this and possibly providing results in a supplement might be helpful.

Lines 178-180 mention that "each document's title and abstract" were used to provide tf-idf features. This could be a bit confusing for some readers since for several of the datasets it seems that the documents in the dataset might not contain titles and abstracts, or at least it's not clear what one would consider as the title and abstract. For example, for the email datasets, what would be considered as the title and abstract for each document? Consider whether there might be a way to rephrase or clarify this.

Update after Author Response: Thank you for the author responses. Answer 2B about the imbalance ratios being updated after each iteration of active learning was interesting. The authors might find it useful to consult this NAACL paper ( https://aclanthology.org/N09-2035.pdf ), which compares the approach of updating the imbalance ratios after each iteration with a modified approach that might work better when active learning is being used.

---

> ### Author Rebuttal · Authors · 2023-08-29
>
> We thank the reviewers for their constructive comments and feedback on our work. We want to address their questions and comments below.
>
> **Questions For The Authors:**
>
>
> * **2A: Is it possible to just use the text classification label predictions to estimate recall and avoid the counting processes and curve fitting entirely? If not, why not? If so, I think it would be beneficial to provide information on how such approaches behave.**
>
> * Yes, it is possible to use the classification only (classify-and-count quantification: CC-Quant), and we have already explored this and have the results, but we did not include results mainly because of the space limit, and the limitations of this approach.
>
> * One limitation of the classification-only approach is that it does not guarantee the confidence level of the results, unlike the counting process approach which estimates probability distribution with a defined confidence level. Another limitation is that it does not model or consider the occurrence rate of relevant documents within the ranking, which is usually decreasing.
>
> * Our experiments on using the classification only as a stopping rule reached lower reliability levels than combining it with the counting process  (i.e. its reliability ranges from 0.96 to 0.80 for CLEF datasets, and 0.93 for RCV1 dataset). Moreover, it also undershoots desired recall for the CLEF2019 dataset on average.
>
> ---
>
>
> * **2B: Lines 198-202 discuss the imbalance ratio and state it is set based on the count of the minority class over the majority class "in training data". Is the training data changing at each iteration of TAR? If so, does this mean that the imbalance ratio can change each iteration after more human-provided relevancy labels are provided? If this is not right, then can you explain how it is set?**
>
> * Yes, this is right. The training data changes at each iteration as more examined documents are included, and thus, the imbalance ratio is also updated and re-calculated based on the majority/minority class distribution of the training dataset at each iteration.
>
> ---
>
> **Typos Grammar Style And Presentation Improvements:**
>
> Thanks a lot, we’ll fix this.
>
> Below are answers to the following raised points:
> * **How are the sample points chosen? For example, are they equally spaced in the relevancy ranking list and systematically chosen, or are they randomly chosen? How many sample points are used? What is the window size W and how is it chosen? Is it fixed or can it vary?**
>
> * sample points are equally spaced across the ranking and selected from within each window, at the same position. The ranking is divided into 10 windows, and the total points == total number of windows == 10.
> * Window size are fixed for each topic, but not fixed across topics, since it varies depending on the topic size (total number of documents), but the total number of windows is fixed for all topics (i.e. 10 windows).
>  * *window size = total number of documents / total number of windows = total number of documents / 10*
>
>
> * **BioBERT and PubMedBERT:**
>
> * BioBERT and PubMedBERT were tested on CLEF datasets only. Other more general BERT models would be more appropriate for TREC Total Recall, TREC Legal, and RCV1 datasets.
>
>
> * **Document Content:**
>
> * The classifier was based on scikit-learn using TF-IDF scores of each document’s content as features, the content is title and abstract for CLEF datasets, email messages for TREC datasets (including title and message body), and news articles content for the RCV1 dataset.
>
> ---
>
> **Other Comments**
> * The main reason behind the counting process not fitting well is that it fits the curve based on the explored portion of the documents only, which may not always be representative of the remaining un-explored documents, and here comes the benefit of adding the classifier to predict the labels of the un-explored documents and fit the curve on the whole ranking. We already have figures that compare real data points vs. fitted rate curves, and we could add them in the extra page space provided.
>
> * We have already experimented using other rate functions e.g. Exponential, which while reaching lower recall levels, the reliability was also lower for some datasets, that is why we reported the Power Law results. Also, regarding comparing the benefit of adding the classifier, they all follow the same patterns i.e. results of the counting process improved with the classifier integration for all rate functions.

---

### Official Review · Reviewer_yCws · 2023-08-04

**Soundness:** 3

**Excitement:**

3: Ambivalent: It has merits (e.g., it reports state-of-the-art results, the idea is nice), but there are key weaknesses (e.g., it describes incremental work), and it can significantly benefit from another round of revision. However, I won't object to accepting it if my co-reviewers champion it.

**Missing References:**

The paper has enough references.

**Paper Topic And Main Contributions:**

The paper proposes a solution for optimizing the stopping rule in Technology Assisted Review. The main aim of the stopping rule is to cut the list of documents returned by the retrieval system as an answer to a query. The challenge here is to find the optimal cut with the best recall and minimum non relevant documents that can be filtered manually. The main contribution of the presented work is integration of the text classifier into the existing approach. The text classifier is trained on the most relevant documents and classifies the rest of the documents in order to select the most relevant and remove non relevant more effectively.
The presented experiments on six widely used datasets demonstrated the reduction of additional non relevant documents with minimum drop in recall.


**Questions For The Authors:**

How it may happened that …relevant documents unexpectedly appear late in the ranking.?
In the fragment: “A set of sample points are extracted from the documents…”, do you mean by point a document?
Why you experimented with three values of desired recall {0.9, 0.8, 0.7} while in the Table with the results Recall always is higher than 0.9?


**Reasons To Accept:**

The paper address an actual problem in information retrieval domain. It proposed a solution and demonstrated its effectivity on six benchmark datasets.
 The paper has proper organization and contains all necessary sections.  The methodology and experiments are described in details.


**Reasons To Reject:**

The problem addressed in the paper is a particular one. It may be important for the Information Retrieval domain; then it should better presented at the specific conference.
 The solution is cumbersome and the improvement is relatively small.


**Reproducibility:**

3: Could reproduce the results with some difficulty. The settings of parameters are underspecified or subjectively determined; the training/evaluation data are not widely available.

**Reviewer Confidence:**

3: Pretty sure, but there's a chance I missed something. Although I have a good feel for this area in general, I did not carefully check the paper's details, e.g., the math, experimental design, or novelty.

**Typos Grammar Style And Presentation Improvements:**

Some sentences are difficult to comprehend. For example,
040  …occurrences of relevant documents are likely to decrease higher in the ranking…
069 …they do not model the rate at which relevant documents have already been observed.
118 …this provides a reliable indication of the rate at which relevant documents occur within the ranking.

---

> ### Author Rebuttal · Authors · 2023-08-29
>
> We thank the reviewers for their constructive comments and feedback on our work. We want to address their questions and comments below.
>
> **Questions For The Authors:**
>
>
> * **1A: How it may happened that …relevant documents unexpectedly appear late in the ranking.?**
>
> * This depends on the ranker and how it perfectly ranked the topic’s relevant documents. For most topics, the ranker placed the relevant documents early in the ranking, but for some topics, the ranker failed to do so, and some of those relevant documents appeared after a big portion of non-relevant documents, which is unexpected for the curve fitting, and hence the classifier predictions would help to spot those documents.
>
> ---
>
> * **1B: In the fragment: “A set of sample points are extracted from the documents…”, do you mean by point a document?**
>
> * Yes, it is a document selected within a specified window of documents, and its relevance judgement is calculated using equation 1. The whole ranking is divided into windows, and a point (document) from each window is selected. This set of points, which is a subset of the complete document collection, is then used to fit the curve.
>
> ---
>
> * **1C: Why you experimented with three values of desired recall {0.9, 0.8, 0.7} while in the Table with the results Recall always is higher than 0.9?**
>
> * Our model goal is to make sure that the desired recall is achieved while minimising cost and maintaining high reliability. Therefore, although it is overshooting, the cost is reduced with lower desired recall levels.
>
> ---
>
> **Typos Grammar Style And Presentation Improvements:**
>
> Thanks a lot, we’ll fix this.
> * 040: most relevant documents appear early in the ranking
> * 069: the classifier alone does not consider or model the occurrence rate at which relevant documents have already been observed
> * 118:  it relies on the assumption that the examined portion of documents provides a reliable indication of the rate at which relevant documents occur within the ranking

---

### Meta-Review · Area_Chair_vfgq · 2023-09-07

**Recommendation:** 3

**Metareview:**

The reviewers share some similar concerns, especially in the lack of traditional baseline comparison. According to the rebuttal information provided by the authors, they have performed related experiments and will add these materials in the extra page if accepted. I therefore believe the authors have answered most of the reviewers' concerns.

---

### Decision · Program_Chairs · 2023-10-07

**Decision:**

Accept-Findings

**Comment:**

The reviewers share some similar concerns, especially in the lack of traditional baseline comparison. According to the rebuttal information provided by the authors, they have performed related experiments and will add these materials in the extra page if accepted. I therefore believe the authors have answered most of the reviewers' concerns.